chemical physics

kaempferol, alkyltrimethylammonium bromide, cohesive force, Stern–Volmer quenching constant, activation energy

**Author for correspondence:**
Ajaya Bhattarai
e-mail: bkajaya@yahoo.com

This article has been edited by the Royal Society of Chemistry, including the commissioning, peer review process and editorial aspects up to the point of acceptance.

# Physico-chemical and spectroscopic investigation of flavonoid dispersed $C_n$TAB micelles

Dileep Kumar[1,2], K. M. Sachin[3,6], Naveen Kumari[4] and Ajaya Bhattarai[3,5]

[1]Division of Computational Physics, Institute for Computational Science, Ton Duc Thang University, Ho Chi Minh City, Vietnam
[2]Faculty of Applied Sciences, Ton Duc Thang University, Ho Chi Minh City, Vietnam
[3]School of Chemical Sciences, Central University of Gujarat, Gandhinagar, India
[4]Department of Chemistry, Deenbandhu Chhotu Ram University of Science and Technology, Murthal, Haryana, India
[5]Department of Chemistry, Tribhuvan University, M.M.A.M. Campus, Biratnagar, Nepal
[6]Department of Chemistry, School of Science, Swarrnim Startup and Innovation University, Gandhinagar, Gujarat, India

DK, 0000-0003-2913-5032; KMS, 0000-0003-3766-2588; NK, 0000-0001-9108-346X; AB, 0000-0002-2648-4686

In this study, kaempferol ($0.2\,\mathrm{m/mmol\,kg^{-1}}$) dispersed cationic surfactant micelles were prepared as a function of alkyltrimethylammonium bromide ($C_n$TAB) hydrophobicity (C = 12 to C = 16). The dispersion study of kaempferol in different $C_n$TAB, i.e. dodecyltrimethylammonium bromide (C = 12), tetradecyltrimethylammonium bromide (C = 14) and hexadecyltrimethylammonium bromide (C = 16), was conducted with the physico-chemical properties of density, sound velocity, viscosity, surface tension, isentropic compressibility, acoustic impedance, surface excess concentration and area occupied per molecule and thermodynamic parameters Gibbs free energy, enthalpy and activation energy measured at 298.15 K. These properties were measured with varying concentration of $C_n$TAB from 0.0260 to $0.0305\,\mathrm{mol\,kg^{-1}}$ in a 10% (w/w) aqueous dimethyl sulfoxide solvent system. The variations in these measured properties have been used to infer the kaempferol dispersion stability via hydrophobic–hydrophilic, hydrophilic–hydrophilic, van der Waals, hydrogen bonding and other non-covalent interactions.

## 1. Introduction

Bio-flavonoids are polyphenolic molecules that are widely present in many food plant origins. Among the several bio-flavonoids,

kaempferol (3,4′,5,7-tetrahydroxyflavone), the major demonstrative subclass of flavonol, is the most useful dietary supplement [1–4]. The kaempferol flavonol is present in onions, apples and tea plants and others with a wide range of health benefits including radical scavenging, anti-cancer, anti-inflammatory and antiviral activities. However, kaempferol is a hydrophobic low water-soluble biomolecule; due to this, it has limited permeability and poor absorption during oral administration [5]. Hence, the solubility and permeability of such flavonol are the key factors for oral bioavailability enhancement [6]. For that, the methods to enhance the solubility and stability of such biomolecules are of great importance for food and drug development [4,7,8].

In recent years, micellar systems have been widely used in formulation development to resolve poor solubility issues [9–12]. These are colloidal micro-/nanodispersion particles that are present in the core–shell of surfactant/polymer or both [13,14]. Hence, these systems are mainly used to deliver biomolecule substances in aqueous or lipid solution form [15,16]. Surfactants are well-known colloidal dispersions which consist of philic and phobic parts. Their specific properties including higher solubility, good stability, perfect micelle formation and non-toxicity represent a wide window of new formulation development [17–23]. With these specific properties, they also act as green solvents and have the potential to replace organic solvents in new drug discovery and development.

The perfect combination of hydrophobic biomolecules, surfactants and steric interactions determines the structural behaviour of biological macromolecules and their nano/micro assemblies' formation [24–26]. The thermodynamics of these nano/micro assemblies or hydrophobic interactions have been studied for many years, as having the philic–phobic dynamics of ionic, electrostatic, van der Waals and the role of water molecules in the interactions [27].

Despite the significant improvement in this research area, many fundamental queries persist unsolved, and it is very important to understand the structural behaviour of hydrophobic molecules with their thermodynamics. The molecular interaction between biomolecules and surfactants (cationic, anionic or non-ionic) in aqueous solution during micelle formation is of great importance to understanding the basic research in the field of surface chemistry and others as well.

We studied the physico-chemical and spectroscopic properties of dispersed flavonoids in the presence of increasing alkyl chain lengths of cationic surfactants. The dispersion is due to non-covalent bonding, such as cohesive force (CF), electrostatic interaction, intermolecular interaction, hydrogen bonding, hydrophobic–hydrophobic, hydrophobic–hydrophilic and ionic interactions. Flavonoids were used in this study because they have anti-oxidant, anti-inflammatory, anti-cancer, anti-tumor and other activities. Surfactants with flavonoids reduce the surface energy and surface tension so that this solution can be used for the preparation of formulations in the pharmaceutical and medicinal fields.

In recent years, it has been reported that a mixture of surfactants as cationic, anionic or non-ionic forms stochiometric complexion [28] which could show precipitation, coalescence or aggregation, while the presence of an excess of the other component may lead to formation of mixed micelles [29,30]. It has also been described that the interaction between opposite charge surfactants with aromatic or aliphatic tails generates higher exothermicity ($\Delta H$) with lower entropy ($\Delta S$) of the system [31,32]. These changes are reflected in the different electrostatic, aliphatic or aromatic chain interactions. However, the physical reason behind these system interactions is still unclear.

There is little literature [33,34] about cationic surfactants with flavonoids. Abbott and Sharma used routine trihydrate as a flavonoid and cetyltrimethylammonium bromide (CTAB) as a cationic surfactant. Micellization parameters were calculated, including standard enthalpy change, standard entropy change and standard Gibbs free energy change. Furthermore, they analysed the nature of interactions and flow properties of the system in terms of density ($\rho$), ultrasonic sound velocity ($u$) and relative viscosity ($\eta$). In addition to calculating various acoustic parameters, they performed FTIR and $^1$H NMR spectroscopic analysis. Kaempferol was not studied with CTAB or dodecyltrimethylammonium bromide (DTAB) or TTAB. Therefore, to understand the thermodynamic variations, we have conducted a physico-chemical dispersion study of kaempferol in aliphatic cationic surfactants. Here, we present the experimental evidence of hydrophobic molecule dispersion via micelle formation with the physico-chemical and thermodynamic measurements.

## 2. Experimental procedure

### 2.1. Materials

Details of chemicals used without any additional purging are mentioned in table 1. Before the study, surfactants were put in a desiccator filled with $P_2O_5$. Kaempferol was securely put in a desiccator at the place since it has light-sensitive properties.

**Table 1.** Details of the chemicals used in the current study.

| chemicals | MW | source | purity (%) | CAS no. |
|---|---|---|---|---|
| kaempferol | 286.24 | Sigma-Aldrich | ≥97.0 | 520-18-3 |
| DMSO | 078.13 | SRL | 99.5 | 67-68-5 |
| DTAB | 308.34 | Sigma-Aldrich | ≥98.0 | 1119-94-4 |
| TDTAB | 336.40 | Alfa-Aesar | 98 | 1119-97-7 |
| HDTAB | 364.46 | Alfa-Aesar | 98 | 57-09-0 |

## 2.2. Preparation methods

Water + DMSO + kaempferol (WDK) arrangement was independently set up by dissolving 0.2 m/mmol kg$^{-1}$ kaempferol independently in 10% (w/w) dimethyl sulfoxide (DMSO)–water mixed solvent media. This arrangement was saved for approximately 1 h at 1000 rpm for mixing. The 0.0260 to 0.0305 mol kg$^{-1}$ C$_n$TAB were included in WDK independently and set on a magnetic stirrer for 20 min at 1000 rpm. The arrangements were set up at room temperature and $p = 0.1$ MPa atmospheric pressure.

## 2.3. Physico-chemical analysis

Densities ($\rho$) and sound velocities of micelles were estimated with an Anton Paar density meter (model 9DSA 5000 M), with ±1 × 10$^{-3}$ K constrained by an underlying Peltier gadget, with ±5 × 10$^{-6}$ g cm$^{-3}$. For an apparatus to be repeatable, it must have a consistent density and sound velocity of ±1 × 10$^{-3}$ kg m$^{-3}$ and ±0.10 m s$^{-1}$. Revealed densities are normal of three repeat estimations with ± 4 × 10$^{-6}$ g cm$^{-3}$ repeatability. Experiments were completed at $T = 298.15$ K with an exactness of ±0.01 K. For every estimation, the cylinder was washed using acetone and dried by going through the U-tube by using a pneumatic machine. Borosil Man Singh Survismeter measured the viscous flow time (VFT) and pendant drop number (PDN) individually. The temperature was maintained by a water bath. After accomplishing an equilibrium, a VFT was noted with a clock of ±0.01 s precision.

We measured PDN with a digital counter. There was an uncertainty of ±2 × 10$^{-6}$ kg m$^{-1}$ s$^{-1}$ in viscosity, and uncertainty of ±0.03 mN m$^{-1}$ in surface tension in each of the three repeat measurements [35,36].

## 2.4. Calculations

Friccohesity of arranged kaempferol micelles was determined using the Man Singh equation (equation (2.1)):

$$\sigma = \frac{\eta}{\gamma_o}\left[\left(\frac{t}{t_o}\right)\left(\frac{n}{n_o}\right)\right], \tag{2.1}$$

where, $\eta_o$, $\gamma_o$, $t_o$, $n_o$ and $\eta$, $\gamma$, $t$, $n$ are the viscosity, surface tension, VFT and PDN of solvent and sample respectively.

Isentropic compressibility of kaempferol micelles was calculated by Laplace–Newton using equation (2.2):

$$k_s = \frac{1}{\rho u^2}. \tag{2.2}$$

Here $\rho$ = density and $u$ = sound velocity for the blank and kaempferol micelles.

The acoustic impedance ($Z$) has been measured using equation (2.3):

$$Z = \rho \cdot u. \tag{2.3}$$

Surface excess concentration ($\Gamma_{max}$) measurements can be determined by using the Gibbs relation (equation (2.4)):

$$\Gamma_{max} = -\frac{1}{2.303nRT}\left(\frac{\partial \gamma}{\partial \mathrm{Log}c}\right)_{T,P}, \tag{2.4}$$

where $T$ is the absolute temperature, $R$ is the gas constant and $\partial\gamma/\partial\mathrm{Log}c$ is the slope of $\gamma$ versus logarithm of oil concentration plot at 298.15 K.

## 2.5. Specific conductance measurement

Conductivity was determined with India-made conductivity method at a frequency of 50 Hz at $T = 298.15$ K using a dipped-type electrode. Temperature (298.15 K) was used to prepare the required solutions.

## 2.6. UV–visible spectroscopy

The absorbance of micelles was recorded with a Spectro 2060 Plus UV–visible spectrometer having a 1 cm path length cuvette made of quartz crystal. The spectral investigation was achieved in different ranges of wavelength varying from 200 to 600 nm at 298.15 K.

## 2.7. Fluorescence study

The fluorometric studies of kaempferol in 0.0260 to 0.0305 mol kg$^{-1}$ C$_n$TAB were performed using a spectrophotometer (Jasco model FP8300) by measuring the excitation $\lambda$ at 363 nm for kaempferol. Emission spectra were noted at the varied ranges of $\lambda = 365$ nm to 600 nm.

## 2.8. Analysis of statistical data

All results are accounted for the standard deviation, standard uncertainties (0.68 degrees of certainty) and consolidated extended uncertainties (0.95 degrees of certainty).

# 3. Results and discussion

## 3.1. Physico-chemical properties estimation model

The $\rho$ and $u$ values of the system determined the packing factor molecular interactions and comparative strength of solute–solvent interactions with internal pressure [37]. Generally, with increasing temperature of solutions, kinetic energy increased via stronger oscillation of molecules which expand the volume of the system with weakening in binding forces [38]. However, in this current study, we studied the molecular arrangement and hydrophobic effects of surfactant on the kaempferol dispersion at ideal room temperature, i.e. 298.15 K. Table 2 shows the $\rho$ and $u$ of kaempferol dispersed in 0.0260 to 0.0305 mol kg$^{-1}$ C$_n$TAB, which demonstrated the packing factor of system with increasing hydrophobicity with varying concentration [39].

   With increasing the 0.0260 to 0.0305 mol kg$^{-1}$ C$_n$TAB at 298.15 K (figure 1), the $\rho$ values increased due to continuously increasing hydrophobicity from C12 to C16 with cohesion is weakened and increases intermolecular forces, frictional forces in the bulk region of the solution. The $\rho$ and $u$ values decreased as DTAB > TDTAB > HDTAB with DMSO quenched 0.2 m mmol kg$^{-1}$ of kaempferol, which is indicating the comparative hydrophobic strength of C$_n$TAB which indicates the intermolecular and electrostatic interaction between solute and solvent systems. The decrease in $\rho$ values and increase in $u$ values for 0.2 m mmol kg$^{-1}$ kaempferol with increasing hydrophobicity identify the dominant hydrophobic–hydrophobic interaction (H$_b$H$_b$I) between kaempferol and alkyl chain of C$_n$TAB. The increase in hydrophobicities (C12–C16) causes stronger molecular oscillations with stronger H$_b$H$_b$I showed the compression of the volume. In another, from C12 to C16, the +I effect releases electron activity of methyl, which reduces the positive charge on quaternary ammonia ($-$N$^+\equiv$) of C$_n$TAB with stronger coulombic force [37].

   Table 3 presents the surface tension ($\gamma$) and viscosity ($\eta$) of the 0.2 m mmol kg$^{-1}$ of kaempferol dispersed in 0.0260–0.0305 mol kg$^{-1}$ C$_n$TAB, where $\gamma$ values are decreased and $\eta$ are increased with increasing concentration 0.0260 to 0.0305 mol kg$^{-1}$ C$_n$TAB at 298.15 K. With DTAB, the $\gamma$ values increased up to 0.0280 mol kg$^{-1}$ by 3.7% and again decreased by 0.52% for 0.0285 mol kg$^{-1}$, and finally constant up to 0.0305 mol kg$^{-1}$. With tetradecyltrimethylammonium bromide (TDTAB), the $\gamma$ values increased from 0.0260 to 0.0275 mol kg$^{-1}$ by 4.3%, decreased from 0.0275 to 0.0280 mol kg$^{-1}$ by

**Table 2.** Density and sound velocity for kaempferol dispersed micelles at 298.15 K and 0.1 MPa (WDK: water + DMSO + kaempferol).

| $m$ (mol kg$^{-1}$) | WDK-DTAB | WDK-TDTAB | WDK-HDTAB |
|---|---|---|---|
| $\rho/10^3$ kg m$^{-3}$ | | | |
| 0.0000 | 1.010807 | 1.010807 | 1.010807 |
| 0.0260 | 1.011225 | 1.010939 | 1.010589 |
| 0.0265 | 1.011268 | 1.010954 | 1.010714 |
| 0.0270 | 1.011284 | 1.010969 | 1.010707 |
| 0.0275 | 1.011295 | 1.010986 | 1.010725 |
| 0.0280 | 1.011301 | 1.01096 | 1.010736 |
| 0.0285 | 1.011300 | 1.010986 | 1.010741 |
| 0.0290 | 1.011305 | 1.010991 | 1.010764 |
| 0.0295 | 1.011312 | 1.011007 | 1.010769 |
| 0.0300 | 1.011319 | 1.011013 | 1.010783 |
| 0.0305 | 1.011323 | 1.011017 | 1.010807 |
| $u$/m s$^{-1}$ | | | |
| 0.0000 | 1548.94 | 1548.94 | 1548.94 |
| 0.0260 | 1552.31 | 1549.70 | 1548.40 |
| 0.0265 | 1552.37 | 1549.72 | 1548.58 |
| 0.0270 | 1552.42 | 1549.83 | 1548.50 |
| 0.0275 | 1552.60 | 1549.92 | 1548.59 |
| 0.0280 | 1552.73 | 1549.86 | 1548.68 |
| 0.0285 | 1552.53 | 1549.92 | 1548.77 |
| 0.0290 | 1552.67 | 1549.94 | 1548.85 |
| 0.0295 | 1552.77 | 1549.96 | 1548.94 |
| 0.0300 | 1552.82 | 1550.03 | 1549.13 |

1.25% and constant up to 0.0305 mol kg$^{-1}$. With hexadecyltrimethylammonium bromide (HDTAB) from 0.0260 to 0.0305 mol kg$^{-1}$ the $\gamma$ values constantly increased by 10.8%.

The $\eta$ values increased from 0.0260 to 0.0305 mol kg$^{-1}$ C$_n$TAB (figure 2), with DTAB, from 0.0260 to 0.0280 mol kg$^{-1}$ decreased by 1.6% and increased by 0.54% with 0.0285 mol kg$^{-1}$, and decreased by 0.82% with 0.0305 mol kg$^{-1}$. With TDTAB, $\eta$ values decreased from 0.0260 to 0.0275 mol kg$^{-1}$ by 0.73% and again increased with 0.0280 mol kg$^{-1}$ by 0.48% and, decreased by 0.95% up to 0.0305 mol kg$^{-1}$. With HDTAB, $\eta$ value decreased from 0.0260 to 0.0265 mol kg$^{-1}$ by 0.22%, increased from 0.0265 to 0.0270 mol kg$^{-1}$ by 0.25% and decreased from 0.0270 to 0.0305 mol kg$^{-1}$ by 1.67%. Hence both $\eta$ and $\gamma$ values in the middle concentration show twisting like increasing and decreasing behaviour which could be indicative of the formation of micellization in the solution and pre- and post-micellization situation of the cationic surfactants in the presence of flavonoid. And another reason may be possible which is that slightly increasing and decreasing values may be due to the electrostatic force of attraction and induced coulombic force generated via hydration spheres of free Br$^-$ ions and DMSO with $-$N$^+\equiv$ ions (figure 3). The increase in the $\gamma$ values from 0.0260 to 0.0305 mol kg$^{-1}$ C$_n$TAB may be due to higher predicted dispersion to electrolytic behaviour of system ingredients.

The increasing trends are defining the role of additional C=C bond in kaempferol, higher hydrophilicity, which shows the increase in $\gamma$ values. In another, delocalized benzene rings of kaempferol have weakly interacting $-$OH groups which also showed weaker van der Waals interaction, intermolecular and electrostatic interaction, with lower $\eta$ values. Hence, the $\eta$ and $\gamma$ of the behaviour of the solution are inversely proportional which supports the good dispersion properties of the solution. Henceforth, in this study, both $\eta$ and $\gamma$ of the 0.2 m mmol kg$^{-1}$ of kaempferol dispersed in 0.0260 to 0.0305 mol kg$^{-1}$ C$_n$TAB were observed as inversely proportional relationships. The decrease in the $\eta$ values indicates the reduction in the comparative interactions [40,41].

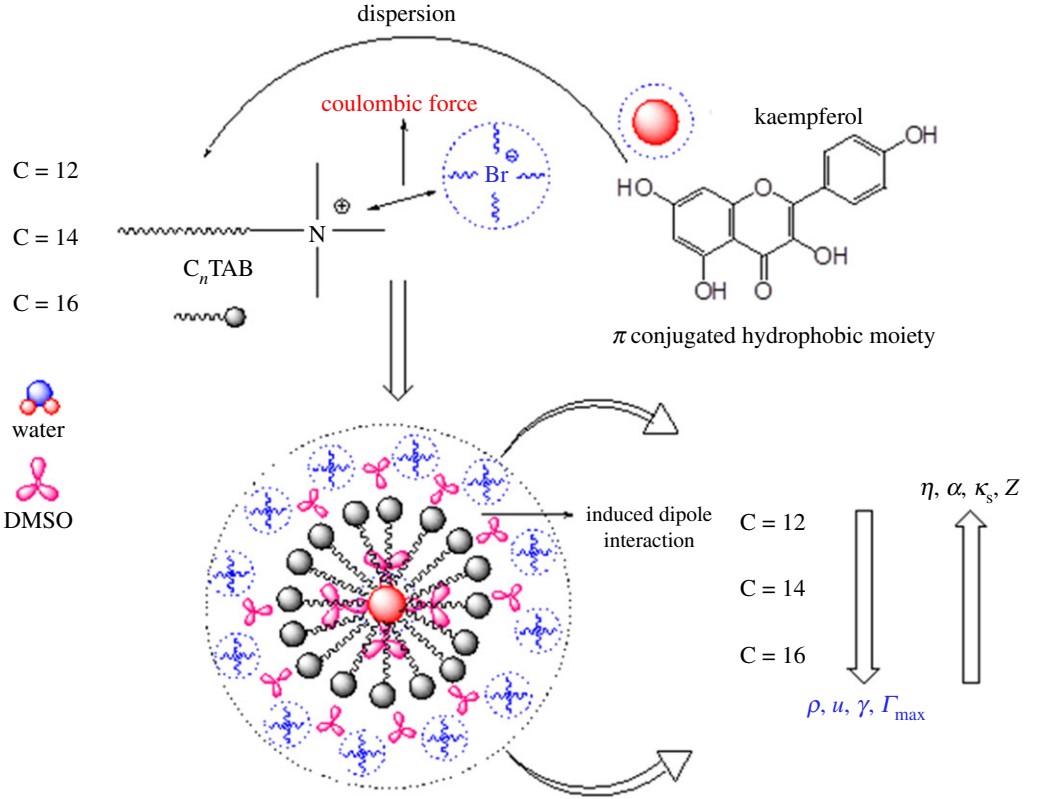

**Figure 1.** The possible micelle formation by $C_n$TAB with 0.2 m mmol kg$^{-1}$ of kaempferol with varying physico-chemical properties.

## 3.2. Surface properties

Surface tension is a measure of the structural activity of molecules at the surface as well as the role of CFs which are present in between solute and solvent systems. In this study, an addition of different concentration of the surfactants decreases the surface energy of the solution. For understating the 0.2 m mmol kg$^{-1}$ of kaempferol dispersion in 0.0260–0.0305 mol kg$^{-1}$ $C_n$TAB, the effect of $\equiv N^+-$, Br$^-$ and DMSO with surface segregation of water and micelle interface, $\Gamma_{max}$ and $1/\Gamma_{max}$ was determined at 298.15 K (figure 4). An area occupied per molecule is indicated in tables 4 and 5 showing the $C_n$TAB (0.0260–0.0305 mol kg$^{-1}$) concentration-dependent $\Gamma_{max}$ and $(\Gamma_{max})^{-1}$ value of 0.2 m mmol kg$^{-1}$ of kaempferol dispersion. $\Gamma_{max}$ values increased from 0.0260 to 0.0305 mol kg$^{-1}$ of $C_n$TAB by 2.6, 3.5 and 3.1% with DTAB, TDATB and HDTAB respectively. Similarly, the $(\Gamma_{max})^{-1}$ value decreased from 0.0260 to 0.0305 mol kg$^{-1}$ of $C_n$TAB by 2.5, 3.2 and 3.3% with DTAB, TDTAB and HDTAB, respectively. These observed $\Gamma_{max}$ trends reflect the continuous increase in hydrophobicity which would have been stimulated via electrostatic interactions [39]. Surface excess concentration represents the total number of molecules present at the surface, and on increasing the concentration of the surfactants in the flavonoid solvent system, due to the London dispersive forces, and van der Waals forces the molecules move towards the surface side with increases the number of molecules at the surface with decreases the area of the molecules.

## 3.3. Friccohesity ($\sigma$): dual force interconversion model

The vibrational molecular dynamic of philic–phobic combination ingredients formed their activities (e.g. vibrational, linear, rotational and translational) at the Lennard-Jones scale. Where these combinations are monitored by the sharing of electron densities on the adjoining atoms via hydrogen bonding, van der Waals forces and London dispersive forces. Solvent–solvent and solute–solvent interactions are facilitated by the spatial arrangements of the molecules [38,42,43]. The $\sigma$ values are fundamentally developed as new sets of combinations via redistribution of electrostatic interaction through increasing hydrophobicity, DMSO and small ionic hydration spheres in electronic supplementary material, figure S1. In similar or dissimilar molecules, the CF is responsible for retaining the integrated structure.

**Table 3.** Viscosity and surface tension for kaempferol dispersed micelles at 298.15 K and 0.1 MPa (WDK: water + DMSO + kaempferol). Molality combined uncertainty $U_c(m)$ of $C_nTAB$ (0.0260–0.0305 mol kg$^{-1}$) in solvents is $\pm2 \times 10^{-4}$ mol kg$^{-1}$. $U_c$ (0.95 confidence level) is $\pm0.34$ mN m$^{-1}$ for surface tension and $\pm5.6 \times 10^{-5}$ kg m$^{-1}$ s$^{-1}$ for viscosity.

| $m$ (mol kg$^{-1}$) | WDK-DTAB | WDK-TDTAB | WDK-HDTAB |
|---|---|---|---|
| $\eta/10^{-3}$ kg m$^{-1}$ s$^{-1}$ | | | |
| 0.0000 | 1.3387 | 1.3387 | 1.3387 |
| 0.0260 | 1.3780 | 1.4388 | 1.9084 |
| 0.0265 | 1.4024 | 1.4606 | 1.9657 |
| 0.0270 | 1.4221 | 1.4801 | 1.9425 |
| 0.0275 | 1.4266 | 1.5031 | 1.9636 |
| 0.0280 | 1.4322 | 1.4846 | 1.9897 |
| 0.0285 | 1.4248 | 1.4918 | 2.0077 |
| 0.0290 | 1.4253 | 1.4923 | 2.0288 |
| 0.0295 | 1.4256 | 1.4931 | 2.0803 |
| 0.0300 | 1.4259 | 1.4933 | 2.1043 |
| 0.0305 | 1.4262 | 1.4937 | 2.1209 |
| $\gamma$/mN m$^{-1}$ | | | |
| 0.0000 | 65.10 | 65.10 | 65.10 |
| 0.0260 | 36.90 | 32.76 | 31.74 |
| 0.0265 | 36.70 | 32.68 | 31.67 |
| 0.0270 | 36.60 | 32.60 | 31.75 |
| 0.0275 | 36.41 | 32.52 | 31.67 |
| 0.0280 | 36.31 | 32.68 | 31.60 |
| 0.0285 | 36.51 | 32.52 | 31.52 |
| 0.0290 | 36.41 | 32.45 | 31.45 |
| 0.0295 | 36.31 | 32.37 | 31.38 |
| 0.0300 | 36.21 | 32.36 | 31.37 |
| 0.0305 | 36.21 | 32.35 | 31.36 |

Kaempferol is sparingly soluble in pure water, as it could not dislocate the CFs of water molecules as well as within the bulk water [37].

In other words, the kaempferol molecule has weaker electrostatic dipole interactions except for aromatic −OH groups and C=C. Hence, DMSO as a universal co-solvent has been used to solubilize kaempferol, which is disrupted in water, the stronger CF disperses. DMSO (>S=O) was triggered by the CFs because it had weak surface energy or surface tension. Table 6 depicts the $\sigma$ values of 0.2 m mmol kg$^{-1}$ of kaempferol dispersed in 0.0260–0.0305 mol kg$^{-1}$ of $C_nTAB$. Where the $\sigma$ values directly vary according to viscosity values, showing directly proportional behaviour. The $\sigma$ values first increased by 5.3, 5.0 and 3.1% from 0.0260 to 0.0280, 0.0260 to 0.0275 and 0.0260 to 0.0265 mol kg$^{-1}$ of DTAB, TDTAB and HDTAB, respectively. The $\sigma$ values again decreased by 1.1, 1.7 and 1.4% from 0.0280 to 0.0285, 0.0275 to 0.0280, and 0.0265 to 0.0270 mol kg$^{-1}$ of DTAB, TDTAB and HDTAB, respectively. Finally, $\sigma$ values again increased by 0.9, 1.6 and 10.5% from 0.0285 to 0.0305, 0.0280 to 0.0305 and 0.0270 to 0.0305 mol kg$^{-1}$ of DTAB, TDTAB and HDTAB, respectively. The observed increasing $\sigma$ values imply the efficient CF interconversion to intermolecular interactions [43,44].

## 3.4. Isentropic compressibility ($\kappa_S$) and acoustic impedance (Z)

The $\kappa_S$ for the 0.2 m mmol kg$^{-1}$ of kaempferol dispersion in the variable concentration of $C_nTAB$ at 298.15 K predicted three-dimensional network structure compression of kaempferol, $C_nTAB$ and

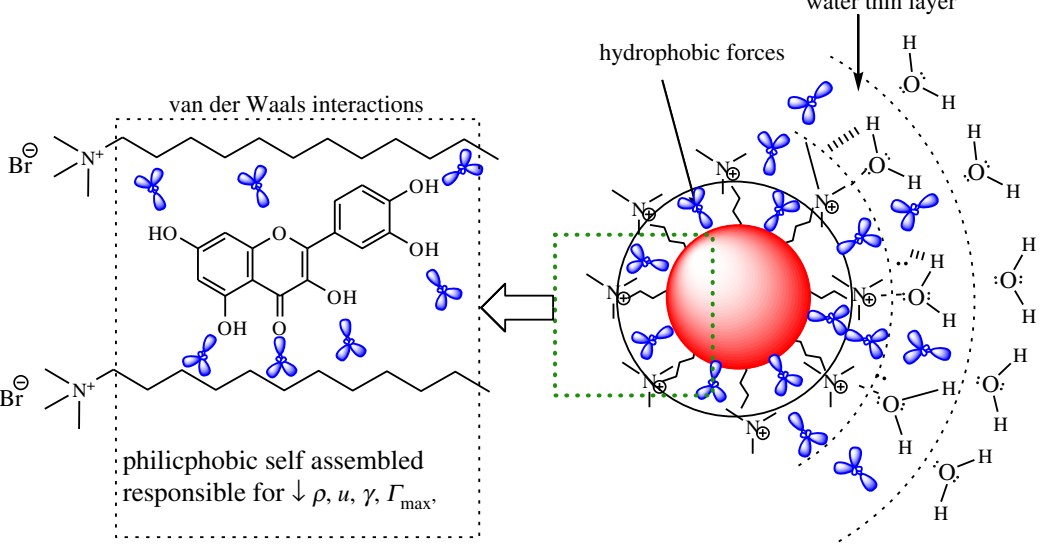

**Figure 2.** Possible interaction mechanism of kaempferol dispersed in C$_n$TAB with DMSO.

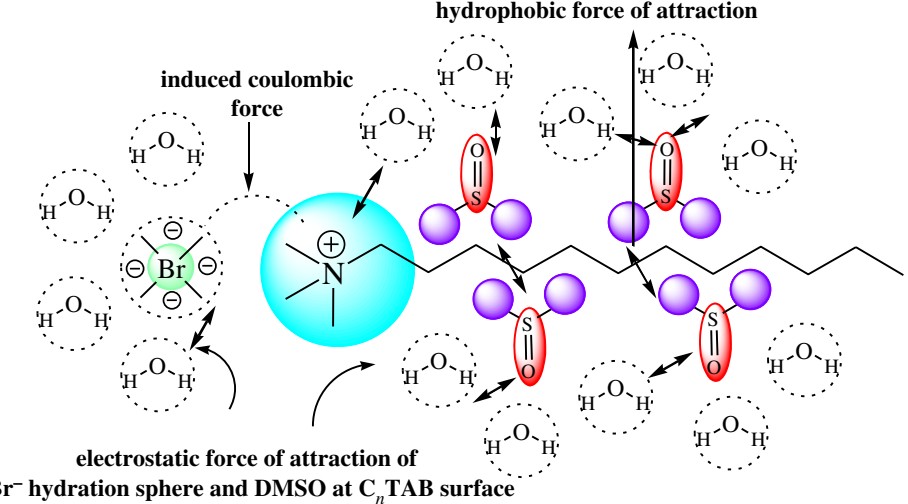

**Figure 3.** Hydration sphere formation and possible interaction mechanism of kaempferol dispersed in C$_n$TAB with DMSO depicted by hydrophobic force gradient via PCP variations.

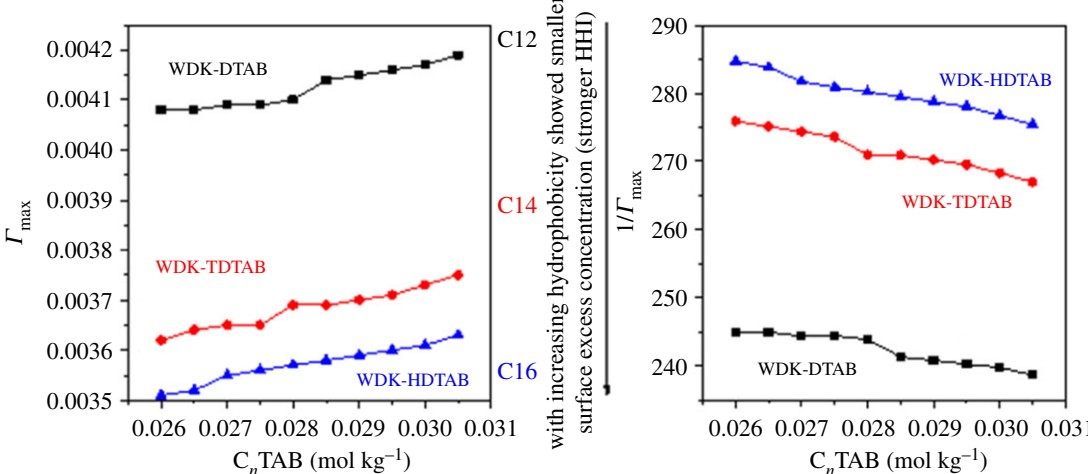

**Figure 4.** Surface excess concentration ($\Gamma_{max}$) and surface area ($1/\Gamma_{max}$) per molecule as a function of hydrophobic strength.

**Table 4.** $\Gamma_{max}$ (mol m$^{-2}$) for kaempferol dispersed micelles at 298.15 K and 0.1 MPa (WDK: water + DMSO + kaempferol). With solvents, the uncertainties in molality U$_c$(m) of C$_n$TAB (0.0260–0.0305 mol kg$^{-1}$) are $\pm 2 \times 10^{-4}$ mol kg$^{-1}$.

| m (mol kg$^{-1}$) | WDK-DTAB | WDK-TDTAB | WDK-HDTAB |
|---|---|---|---|
| 0.0260 | 0.00410 | 0.00364 | 0.00353 |
| 0.0265 | 0.00410 | 0.00366 | 0.00354 |
| 0.0270 | 0.00411 | 0.00367 | 0.00357 |
| 0.0275 | 0.00411 | 0.00367 | 0.00358 |
| 0.0280 | 0.00412 | 0.00371 | 0.00359 |
| 0.0285 | 0.00416 | 0.00371 | 0.00360 |
| 0.0290 | 0.00417 | 0.00372 | 0.00361 |
| 0.0295 | 0.00418 | 0.00373 | 0.00362 |
| 0.0300 | 0.00419 | 0.00375 | 0.00363 |
| 0.0305 | 0.00421 | 0.00377 | 0.00367 |

**Table 5.** $A_{min}$ (m$^2$ mol$^{-1}$) for kaempferol dispersed micelles at 298.15 K and 0.1 MPa (WDK: water + DMSO + kaempferol). With solvents, the uncertainties in molality U$_c$(m) of C$_n$TAB (0.0260–0.0305 mol kg$^{-1}$) are $\pm 2 \times 10^{-4}$ mol kg$^{-1}$.

| m (mol kg$^{-1}$) | WDK-DTAB | WDK-TDTAB | WDK-HDTAB |
|---|---|---|---|
| 0.0260 | 244.90 | 275.85 | 284.67 |
| 0.0265 | 244.95 | 275.07 | 283.82 |
| 0.0270 | 244.35 | 274.32 | 281.70 |
| 0.0275 | 244.43 | 273.58 | 280.92 |
| 0.0280 | 243.86 | 270.89 | 280.17 |
| 0.0285 | 241.34 | 270.86 | 279.43 |
| 0.0290 | 240.81 | 270.19 | 278.71 |
| 0.0295 | 240.29 | 269.52 | 278.01 |
| 0.0300 | 239.79 | 268.24 | 276.68 |
| 0.0305 | 238.66 | 266.97 | 275.37 |

**Table 6.** Friccohesity ($\sigma$/s cm$^{-1}$) for kaempferol dispersed micelles at 298.15 K and 0.1 MPa (WDK: water + DMSO + kaempferol). With solvents, the uncertainties in molality U$_c$(m) of C$_n$TAB (0.0260 to 0.0305 mol kg$^{-1}$) are $\pm 2 \times 10^{-4}$ mol kg$^{-1}$.

| m (mol kg$^{-1}$) | WDK-DTAB | WDK-TDTAB | WDK-HDTAB |
|---|---|---|---|
| 0.0000 | 0.02054 | 0.02054 | 0.02054 |
| 0.0260 | 0.03729 | 0.04385 | 0.06003 |
| 0.0265 | 0.03815 | 0.04462 | 0.06197 |
| 0.0270 | 0.03879 | 0.04533 | 0.06110 |
| 0.0275 | 0.03913 | 0.04615 | 0.06191 |
| 0.0280 | 0.03939 | 0.04536 | 0.06287 |
| 0.0285 | 0.03897 | 0.04580 | 0.06359 |
| 0.0290 | 0.03909 | 0.04593 | 0.06441 |
| 0.0295 | 0.03921 | 0.04606 | 0.06620 |
| 0.0300 | 0.03932 | 0.04607 | 0.06696 |
| 0.0305 | 0.03933 | 0.04608 | 0.06749 |

**Table 7.** Isentropic compressibility ($S_\kappa/10^{12}$ m$^4$ kg s$^2$ mol$^{-2}$) and acoustic impedance ($Z$/g cm$^{-2}$ s$^{-1}$) for kaempferol dispersed micelles at 298.15 K and 0.1 MPa (WDK: water + DMSO + kaempferol). With solvents, the uncertainties in molality U$_c$(m) of C$_n$TAB (0.0260–0.0305 mol kg$^{-1}$) are $\pm 2 \times 10^{-4}$ mol kg$^{-1}$.

| m (mol kg$^{-1}$) | WDK-DTAB | WDK-TDTAB | WDK-HDTAB |
|---|---|---|---|
| $S_\kappa/10^{12}$ m$^4$ kg s$^2$ mol$^{-2}$ | | | |
| 0.0260 | $4.10 \times 10^{-7}$ | $4.12 \times 10^{-7}$ | $4.13 \times 10^{-7}$ |
| 0.0265 | $4.10 \times 10^{-7}$ | $4.12 \times 10^{-7}$ | $4.13 \times 10^{-7}$ |
| 0.0270 | $4.10 \times 10^{-7}$ | $4.12 \times 10^{-7}$ | $4.13 \times 10^{-7}$ |
| 0.0275 | $4.10 \times 10^{-7}$ | $4.12 \times 10^{-7}$ | $4.13 \times 10^{-7}$ |
| 0.0280 | $4.10 \times 10^{-7}$ | $4.12 \times 10^{-7}$ | $4.13 \times 10^{-7}$ |
| 0.0285 | $4.10 \times 10^{-7}$ | $4.12 \times 10^{-7}$ | $4.12 \times 10^{-7}$ |
| 0.0290 | $4.10 \times 10^{-7}$ | $4.12 \times 10^{-7}$ | $4.12 \times 10^{-7}$ |
| 0.0295 | $4.10 \times 10^{-7}$ | $4.12 \times 10^{-7}$ | $4.12 \times 10^{-7}$ |
| 0.0300 | $4.10 \times 10^{-7}$ | $4.12 \times 10^{-7}$ | $4.12 \times 10^{-7}$ |
| 0.0305 | $4.10 \times 10^{-7}$ | $4.12 \times 10^{-7}$ | $4.12 \times 10^{-7}$ |
| $Z$/g cm$^{-2}$ s$^{-1}$ | | | |
| 0.0260 | 1569.73 | 1566.64 | 1564.80 |
| 0.0265 | 1569.87 | 1566.71 | 1565.18 |
| 0.0270 | 1569.95 | 1566.84 | 1565.09 |
| 0.0275 | 1570.13 | 1566.95 | 1565.21 |
| 0.0280 | 1570.28 | 1566.87 | 1565.31 |
| 0.0285 | 1570.08 | 1566.95 | 1565.41 |
| 0.0290 | 1570.23 | 1567.00 | 1565.52 |
| 0.0295 | 1570.34 | 1567.02 | 1565.62 |
| 0.0300 | 1570.40 | 1567.11 | 1565.84 |
| 0.0305 | 1570.45 | 1567.17 | 1565.96 |

DMSO and negative compression effect due to the micelle compactness. The $\kappa_S$ factor significantly determined the magnitude of micelle compactness as well as physico-chemical behaviour of the systems which mainly depends on the hydrogen bonding, molecular arrangement, or complex formations [45]. Similarly, close packing between dispersed systems and the internal pressure in the micellar systems are also determined by $Z$ values. The $\kappa_S$ for the 0.2 m mmol kg$^{-1}$ of kaempferol dispersion in C$_n$TAB is represented in table 7, where the $\kappa_S$ values increased with increasing hydrophobicity. The $\kappa_S$ values increased by approximately $0.381 \pm 0.013$ and $0.171 \pm 0.024$% with TDTAB and HDTAB addition. Hence, the $\kappa_S$ values were observed as HDTAB > TDTAB > DTAB. The $\kappa_S$ values decreased with increasing concentration from 0.0260 to 0.0305 mol kg$^{-1}$ of C$_n$TAB by 0.081, 0.059 and 0.128% with DTAB, TDTAB and HDTAB, respectively. Similarly, table 7 depicts the variation in $Z$ values for 0.2 m mmol kg$^{-1}$ of kaempferol dispersion in C$_n$TAB, where the $Z$ values are decreased with increasing hydrophobicity from C12 to C16. The $Z$ values were obtained as DTAB > TDTAB > HDTAB. Table 7 shows that the $Z$ values are increased from 0.0260 to 0.0305 mol kg$^{-1}$ of C$_n$TAB addition by 0.046, 0.034 and 0.075% with DTAB, TDTAB and HDTAB respectively. The decrease in $\kappa_S$ values indicates the strength of the hydrophobic force of attraction and intermolecular forces, which led to generating phobic–phobic interaction behaviour between kaempferol and C$_n$TAB molecules.

The presence of a longer hydrocarbon chain (from C12 to C16) of C$_n$TAB may have caused structural reorientation and quenching oscillation with DMSO in the medium. Hence, the $\kappa_S$ measurements imply the stability of kaempferol in C$_n$TAB with the development of a new cross-linked hydrophobic interaction in electronic supplementary material, figure S2. The smaller increment in $Z$ values of kaempferol dispersion showed the dominance of hydrophobic interactions. The higher impedance variation generated the high energy that reflected the surface interface of kaempferol, DMSO and C$_n$TAB.

**Table 8.** Conductivity ($\kappa$/mS cm$^{-1}$) for kaempferol dispersed micelles at 298.15 K and 0.1 MPa (WDK: water + DMSO + kaempferol). There are $\pm 2 \times 10^{-4}$ mol kg$^{-1}$ uncertainties in molality $U_c(m)$ in $C_n$TAB (0.0260 to 0.0305 mol kg$^{-1}$) with solvents. The combined expanded uncertainty $U_c$ (0.95 confidence level) is $U_c(\kappa) = \pm 0.60$ mS cm$^{-1}$.

| $m$ (mol kg$^{-1}$) | WDK-DTAB | WDK-TDTAB | WDK-HDTAB |
|---|---|---|---|
| 0.0260 | 1.2177 | 0.6207 | 0.4451 |
| 0.0265 | 1.2189 | 0.6249 | 0.4401 |
| 0.0270 | 1.2203 | 0.6265 | 0.4535 |
| 0.0275 | 1.2232 | 0.6341 | 0.4441 |
| 0.0280 | 1.2235 | 0.6521 | 0.4451 |
| 0.0285 | 1.2401 | 0.6476 | 0.4544 |
| 0.0290 | 1.2299 | 0.6473 | 0.4931 |
| 0.0295 | 1.2454 | 0.6639 | 0.4829 |
| 0.0300 | 1.2746 | 0.6575 | 0.4828 |
| 0.0305 | 1.2755 | 0.6559 | 0.4848 |

Hence, the observed $Z$ values from 0.0260 to 0.0305 mol kg$^{-1}$ of $C_n$TAB could be efficient for unique kaempferol capturing with direct assessment of its release behaviour or therapeutic efficacy. Also, increasing $Z$ values described the higher compressibility that pushes the molecules more tightly to each other by lessening their interaction bond lengths [38,42].

## 3.5. Conductivity measurements ($\kappa$)

The conductivity value indicates the whole population of ions in the solution; it also depends on the mobility, size of the ions and hydration sphere size as well. In this study, we studied the variable concentrations of $C_n$TAB with increasing hydrophobicity. $C_n$TAB has the same ionic moiety in its molecular structure, and on increasing the concentration of $C_n$TAB, the number of ionic moieties increases with increased conductivity value of the solution. The $\kappa$ values of 0.2 m mmol kg$^{-1}$ of kaempferol dispersion in $C_n$TAB from 0.0260 to 0.0305 mol kg$^{-1}$ given in table 8. The $\kappa$ values are increased with increasing concentration from 0.0260 to 0.0305 mol kg$^{-1}$ of $C_n$TAB by 4.7, 5.7 and 8.9% with DTAB, TDTAB and HDTAB respectively. The highest $\kappa$ values were observed with DTAB and lowest with HDTAB, due to increase in hydrophobicity from C12 to C16 in electronic supplementary material, figure S3. The conductivity value order showed the molecular structural behaviour of the surfactants toward the solvent system. In this study, DTAB has the smallest size compared to TDTAB and HDTAB. So, it could form the smallest sized hydration ionic sphere having higher mobility than TDTAB and HDTAB surfactants.

The $\kappa$ value trend was observed as DTAB > TDTAB > HDTAB which is similar to the surface tension trend. Thus, the observed trends imply the increasing binding affinity induced by the hydrophobic force gradient of $C_n$TAB and kaempferol. The kaempferol molecule has C=C bond in the heterocyclic ring with extended $\pi$ conjugation and H$^+$ moiety which approached the DMSO and water molecules to form the acid–base complex. These complexes facilitate the transport of H$^+$ ions in the bulk systems for the conduction theory [36].

## 3.6. Thermodynamic properties

The $\Delta G$, $\Delta H$, $E^*$ and $\Delta S$ values (table 9) are calculated for observing the kaempferol dispersion, patterns of physical and thermodynamic distribution in $C_n$TAB with DMSO. The $E^*$ value is determined by the Arrhenius equation at $T = 298.15$ K and fitted as

$$\log(E) = \log A - \frac{E^*}{2.303RT} \tag{3.1}$$

**Table 9.** Thermodynamic parameters for kaempferol with $C_n$TAB interaction. With solvents, the molality uncertainty $U_c(m)$ of $C_n$TAB (0.0260 to 0.0305 mol kg$^{-1}$) is $\pm 2 \times 10^{-4}$ mol kg$^{-1}$.

| $m$ (mol kg$^{-1}$) | $\Delta S$ (kJ mol$^{-1}$) | $E^*$ (J mol$^{-1}$) | $\Delta H$ (J mol$^{-1}$) | $\Delta G$ (J mol$^{-1}$) |
|---|---|---|---|---|
| DTAB, C = 12 | | | | |
| 0.000 | −19.15 | −249.79 | −5958.51 | −249.79 |
| 0.026 | | −183.91 | −5892.63 | −183.91 |
| 0.027 | | −247.54 | −5956.26 | −247.54 |
| 0.028 | | −179.30 | −5888.02 | −179.30 |
| 0.029 | | −252.03 | −5960.75 | −252.03 |
| 0.030 | | −243.05 | −5951.77 | −243.05 |
| TDTAB, C = 14 | | | | |
| 0.000 | −19.15 | −249.79 | −5958.51 | −249.79 |
| 0.026 | | −225.00 | −5933.72 | −225.00 |
| 0.027 | | −85.29 | −5794.01 | −085.29 |
| 0.028 | | −128.04 | −5836.76 | −128.04 |
| 0.029 | | −087.68 | −5796.40 | −87.68 |
| 0.030 | | 336.73 | −5371.99 | 336.73 |
| HDTAB, C = 16 | | | | |
| 0.000 | −19.15 | −249.79 | −5958.51 | −249.79 |
| 0.026 | | 397.10 | −5311.62 | 397.10 |
| 0.027 | | 414.62 | −5294.10 | 414.62 |
| 0.028 | | 510.22 | −5198.5 | 510.22 |
| 0.029 | | 458.98 | −5249.75 | 458.98 |
| 0.030 | | 441.14 | −5267.58 | 441.14 |

and

$$= \log A - \frac{E^*}{2.303R} \frac{1}{T}, \tag{3.2}$$

where *abs* is the absorbance at 260 nm, $T$ is in kelvin, $R$ is 8.314 J mol$^{-1}$ K, $E^*$ is the activation energy (J mol$^{-1}$) and $A$ is the frequency factor.

Further, $E^*$ values are applied to measure the $\Delta H$ (equation (3.3)) for the pristine and annealed processes:

$$\Delta H = E^* - 2.303RT. \tag{3.3}$$

So, the $\Delta G$ and $\Delta S$ (equations (3.4)–(3.6)) for stable systems are calculated by using equations given as

$$\Delta S = \frac{(E^* - 2.303RT + 2.303RT \ \log(abs))}{T}, \tag{3.4}$$

$$\Delta S = \left(\frac{E^*}{T}\right) - 2.303R[1 - \log(abs)] \tag{3.5}$$

and

$$\Delta G = -2.303RT \ \log(abs). \tag{3.6}$$

Table 9 contains the $E^*$, $\Delta H$, $\Delta S$ and $\Delta G$ for 0.2 m mmol kg$^{-1}$ of kaempferol dispersion in $C_n$TAB from 0.0260 to 0.0305 mol kg$^{-1}$ at 298.15 K implying a kaempferol mono dispersion in $C_n$TAB with DMSO. As $\Delta G$ greater than 0 supported the mono dispersion or nanomicelle development, HB water via $C_n$TAB and DMSO cannot be disrupted because of its thermodynamic incompatibility. Table 9 shows the $\Delta G$ of kaempferol dispersed HDTAB micelles, where the $\Delta G$ greater than 0 is observed due to molecular oscillation inducing the local molecular interactions in the bulk which could affect the thermodynamic and kinetic stability of kaempferol dispersed micelles with hydrophobic

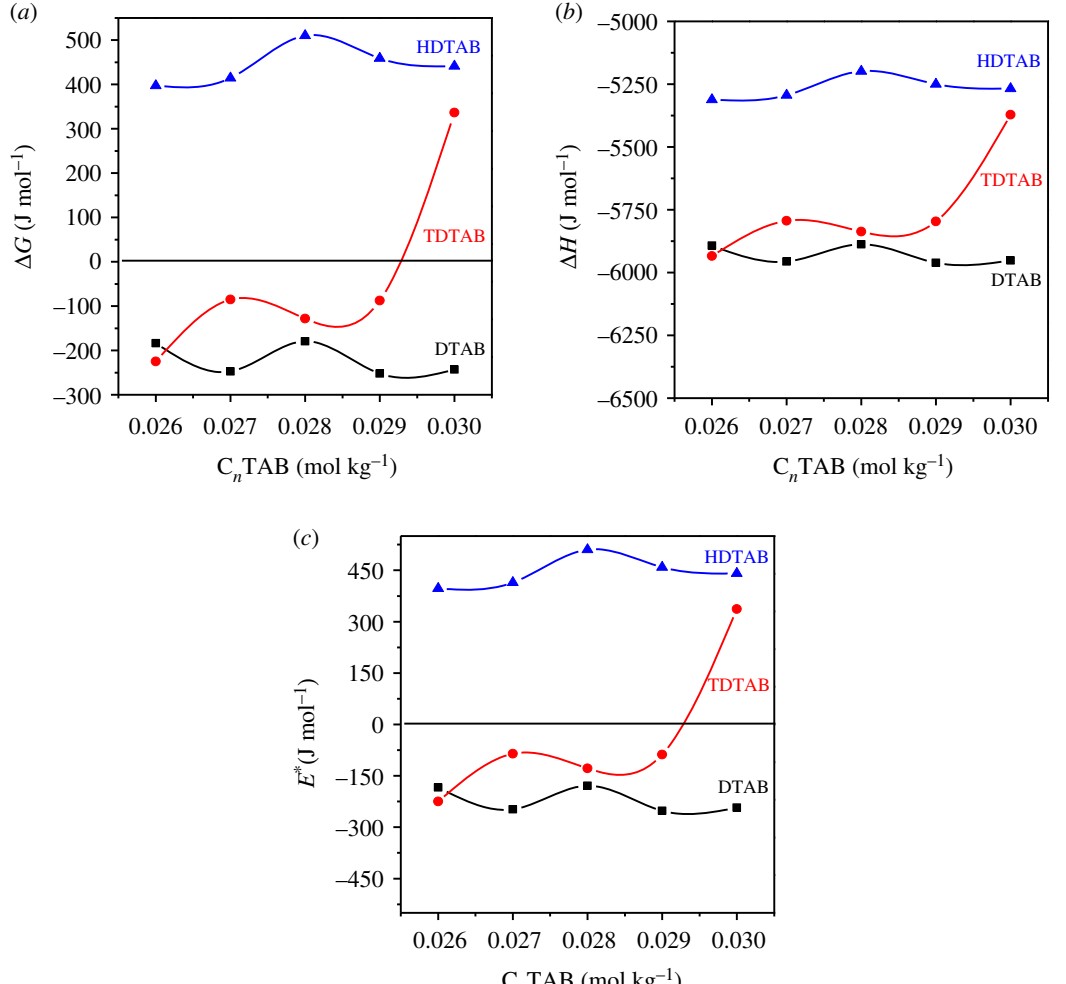

**Figure 5.** Thermodynamic parameters of kaempferol micelles: (*a*) free energy ($\Delta G$), (*b*) enthalpy ($\Delta H$) and (*c*) activation energy ($E^*$).

dominance. In another, non-spontaneous kaempferol micellar formation could be able to produce the increment in medium energy in aggregation or DMSO and Br⁻ via hydration sphere formation. The kaempferol micelles with DTAB and TDTAB produced $\Delta G$ less than 0 at 298.15 K. These more negative values imply the stronger hydrophobic interaction in between C12 and C14 with increasing kinetic energy (figure 5*a*).

Kaempferol dispersion efficiencies are dependent on $\Delta H$ to determine the thermal energy required. $C_n$TAB micelles had negative $\Delta H$ values as the increasing hydrophobicity assisted a thermodynamically stable dispersion of kaempferol via energy release in DTAB > TDATB > HDTAB (figure 5*b*). As a result, the $C_n$TAB with water cannot break the CF of the clustered molecules of kaempferol. Though, $\Delta H$ greater than 0 for DTAB indicates a disturbance of kaempferol CF and releases the excess potential energy to the micelle formation. For determining the interaction kinetics of kaempferol and $C_n$TAB in different molality ratios the activation energies $E^*$ were evaluated. $E^*$ greater than 0 was obtained with HDTAB, with a homogeneous dispersion for kaempferol (figure 5*c*). Hence the observed trends of $E^*$ are supported by $\Delta H$ and $\Delta G$ values. The lower $E^*$ depicted a decrease in the energy barrier of kaempferol during $C_n$TAB dispersion with comparatively weaker interaction and mono-dispersion in the medium [39,43].

## 3.7. Fluorescence mechanism: Stern–Volmer quenching

Stern–Volmer relation is usually employed to discuss fluorescence quenching. In the current study fluorescence quenching was calculated from the following equation:

$$\frac{F_0}{F} = 1 + K_q \tau_0 \, [Q] = 1 + S_{sv} \, [Q], \tag{3.7}$$

**Table 10.** The Stern–Volmer quenching constant ($K_{sv}$), quenching rate constant ($K_q$), binding constant ($K_b$), binding number ($n$) and Gibbs free energy ($\Delta G$) for the interaction of kaempferol with DTAB, TDTAB and HDTAB (measured at 363 nm).

| system | $K_q$ ($10^{11}$ M$^{-1}$ s$^{-1}$) | $\Delta G$ (kJ mol$^{-1}$) | $K_{sv}$ (M$^{-1}$) | $K_b$ (M$^{-1}$) | $n$ |
|---|---|---|---|---|---|
| kaempferol–DTAB | 2.49 | −15.41 | 2.5 | $2.70 \times 10^2$ | 2.28 |
| kaempferol–TDTAB | 0.29 | −15.41 | 0.3 | $2.70 \times 10^2$ | 2.38 |
| kaempferol–HDTAB | 0.36 | −15.41 | 0.4 | $2.70 \times 10^2$ | 2.16 |

where $F_0$ and $F$ are the fluorescence intensities before and after quenching, respectively, $K_q$ is the quenching constant and [Q] is the concentration of quencher.

Stern–Volmer quenching constant $S_{sv}$ is a measure of quenching efficiency and $\tau_0$ is the average lifetime of the biomolecule. $K_q$ is determined by equation (3.8):

$$K_{sv} = K_q \tau_0, \tag{3.8}$$

where $\tau_0$ is $10^{-8}$ s and $F_0/F$ versus [Q] is linearly regressed as $S_{sv}$. Using Stern–Volmer analysis of the relative fluorescence intensity ($F_0/F$) as a function of the quencher concentration [Q], C$_n$TAB explained the quenching mechanism of kaempferol by fluorescence quenching.

Number of binding sites and fluorescence binding constant are obtained from the following equation:

$$\log\left[\frac{F_0 - F}{F}\right] = \log K_b + n \log [Q]. \tag{3.9}$$

A plot of $\log[F_0 - F/F]$ versus $\log[Q]$ is a linear variation with $\log K_b$ acting as intercept and $n$ as slope. Kaempferol affinity for C$_n$TAB is measured by $K_b$, where $n$ is the number of binding sites.

It has been reported that surfactant concentrations in different micellar systems and organic probe that is distributed into the micellar core affect the fluorescence quenching which highly favours the formation of stable micelles. The increase in quenching for the micellar system is mainly due to kinetic and thermodynamic points of view. Table 10 shows $K_{sv}$, $K_q$, $K_b$ and $n$ values for 0.0002 mol kg$^{-1}$ of kaempferol dispersion in C$_n$TAB from 0.0260 to 0.0305 mol kg$^{-1}$. These consequences suggest that C$_n$TAB micelle-enhanced fluorescence quenching by kaempferol is an analytical process rather than a fundamental application.

The interaction of apigenin with bovine serum albumin was studied and reported by spectroscopy of fluorescence and UV–visible absorption. DTAB, one of the similar surfactants in our study, was used to examine the interactions between apigenin and bovine serum albumin. During the binding interaction, van der Waals force and hydrogen bonds played a major role in quenching the bovine serum albumin induced by apigenin [46]. This fulfilled the major gaps in both existing experimental work and the current research.

## 4. Conclusion

Stable kaempferol micelles were successfully made and studied via physico-chemical and thermodynamic measurements through hydrophobic–hydrophobic, ion–hydrophobic/hydrophilic and van der Waals forces impact on dispersion using 10% (w/w) DMSO. The strong HHI between C$_n$TAB (C12–C16) and kaempferol formed structural compactness, evidenced by the lower $\rho$, u, $\gamma$ and $\Gamma_{max}$ where the DMSO, Br− and =N$^+$= groups established a dynamic structural modulation in micelles. The decrease in $\gamma$ values for micelles has indicated the surface activity is effective and the distribution activity is stronger with the increment in $\eta$ values. The decrease in $\Gamma_{max}$ values with higher hydrophobicity less assertive C$_n$TAB molecules to surface with less Brownian motion. We have found an article [46] on the effects of sodium dodecylsulfate and DTAB on the interaction of apigenin and bovine serum albumin which motivated us to study it in the current work. Therefore, in this study, C$_n$TAB ranging from C = 12 to C = 16 has lowered CF and favoured stable micelle development with least $\Delta G$ values. Hydrophobicity and non-covalent bonding have also been determined to have a role in the preparation of flavonoid formulations that can be used in pharmaceutical, biomedical and several other applications.

Data accessibility. Data that support this study have been uploaded as electronic supplementary material.

Authors' contributions. D.K.: Conceptualization, data curation, formal analysis, methodology; K.M.S.: Funding acquisition, project administration, writing—original draft; N.K.: Resources, software, validation; A.B.: Supervision, visualization, writing—review and editing. All authors gave final approval for publication and agreed to be held accountable for the work performed therein.

Competing interests. The authors declare no competing interests.

Funding. We received no funding for this study.

Acknowledgements. Special thanks go to the Central University of Gujarat for infrastructure, and experimental facilities support.

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
