## [Peer Review File · Royal Society Open Science]

Review History

RSOS-210758.R0 (Original submission)

Review form: Reviewer 1 (Prof. Dr. Bidyut Saha)

Is the manuscript scientifically sound in its present form?

Yes

Are the interpretations and conclusions justified by the results?

Yes

Is the language acceptable?

Yes

Do you have any ethical concerns with this paper?

No

Have you any concerns about statistical analyses in this paper?

No

Recommendation?

Accept with minor revision (please list in comments)

Comments to the Author(s)

Minor revision

Review form: Reviewer 2**Is the manuscript scientifically sound in its present form?**

Yes

Are the interpretations and conclusions justified by the results?

Yes

Is the language acceptable?

Yes

Do you have any ethical concerns with this paper?

No

Have you any concerns about statistical analyses in this paper?

Yes

Recommendation?

Major revision is needed (please make suggestions in comments)

Comments to the Author(s)

Manuscript ID: RSOS-210758

Title: Physicochemical and spectroscopic investigation of flavonoid dispersed CnTAB micelles

This manuscript has some concerns presented below. However, this work may be recommended to publish after careful revision considering the below points and the revised version should be reviewed again before the publication in Royal Society Open Science:

- (i) Authors are optimistic in the estimation of standard uncertainty in measured and derived properties. Should be re-evaluated. The uncertainty of each measured variable must be given and combined in quadrature to give the uncertainty in the derived properties, which must be stated at a defined confidence interval. The number of significant digits must not exceed one more digit than specified by the estimated uncertainty, especially derived and excess properties. The purity of the sample must be considered in the estimates of uncertainty. Considering the revised uncertainty, all data should be presented in appropriate significant numbers.
- (ii) Comment on the effect of purity of compound on studied properties of flavonoid dispersed CnTAB micelles.
- (iii) Authors focused more on explaining the trends observed rather than coming up with convincing explanations/ arguments. Still, there is scope to improve the discussion part by elaborating the effect/role of flavonoid /surfactant/ composition solute/solvent for the studied properties of systems considered in this work.
- (iv) Discussion and Conclusions should be elaborated further along with a comparison of results for similar surfactants and efforts to be made towards evidencing major gaps in both existing experimental work/ theoretical & computational models.

(v) Further justification/ explanation is essential for most of the claims about the hydrophobic-hydrophilic, hydrophilic-hydrophilic, Van der Waals and hydrogen, bonding, and other non-covalent interactions are mainly based on derivatives and double derivatives of derived data associated with substantial uncertainties.

Decision letter (RSOS-210758.R0)

Dear Dr Bhattarai:

Title: Physicochemical and spectroscopic investigation of flavonoid dispersed CnTAB micelles
Manuscript ID: RSOS-210758

The editor assigned to your manuscript has now received comments from reviewers. We would like you to revise your paper in accordance with the referee and Subject Editor suggestions which can be found below (not including confidential reports to the Editor). Please note this decision does not guarantee eventual acceptance.

Please submit your revised paper before 27-Oct-2021. Please note that the revision deadline will expire at 00.00am on this date. If we do not hear from you within this time then it will be assumed that the paper has been withdrawn. In exceptional circumstances, extensions may be possible if agreed with the Editorial Office in advance. We do not allow multiple rounds of revision so we urge you to make every effort to fully address all of the comments at this stage. If deemed necessary by the Editors, your manuscript will be sent back to one or more of the original reviewers for assessment. If the original reviewers are not available we may invite new reviewers.

Yours sincerely,
Dr Ellis Wilde

Publishing Editor, Journals

On behalf of the Subject Editor Professor Anthony Stace and the Associate Editor Dr Debashree Ghosh.

RSC Associate Editor

Comments to the Author:

The manuscript requires major revision and the authors should provide a point-by-point reply to the concerns raised by the referees.

RSC Subject Editor

Comments to the Author:

(There are no comments.)

Reviewers' Comments to Author:

Reviewer: 1

Comments to the Author(s)

Minor revision

In addition to references 9, 10, use these also

J. Mol. Liq. 310 (2020) 113224

J. Mol. Liq. 293 (2019) 111475

1. Insert a list of abbreviation used.
2. Add one or two Keywords in the revised text.
3. The authors should highlight the importance of their work in the introduction.
4. The length of the manuscript is somehow sufficient. In my opinion, it should more elaborate the introduction part to meet the required standard.
5. What is the novelty of this work, explain nicely.
6. All the equation numbers must be mentioned in the revised text.
7. Give only exact findings from the results and discussion in the conclusion section.
8. Reference section must be taken into consideration. Check the references and correct them according to the journal style and format.
9. English language needs to be improved throughout the manuscript.

Reviewer: 2

Comments to the Author(s)

Manuscript ID: RSOS-210758

Title: Physicochemical and spectroscopic investigation of flavonoid dispersed CnTAB micelles

This manuscript has some concerns presented below. However, this work may be recommended to publish after careful revision considering the below points and the revised version should be reviewed again before the publication in Royal Society Open Science:

(i) Authors are optimistic in the estimation of standard uncertainty in measured and derived properties. Should be re-evaluated. The uncertainty of each measured variable must be given and combined in quadrature to give the uncertainty in the derived properties, which must be stated at a defined confidence interval. The number of significant digits must not exceed one more digit than specified by the estimated uncertainty, especially derived and excess properties. The purity of the sample must be considered in the estimates of uncertainty. Considering the revised uncertainty, all data should be presented in appropriate significant numbers.

(ii) Comment on the effect of purity of compound on studied properties of flavonoid dispersed CnTAB micelles.

(iii) Authors focused more on explaining the trends observed rather than coming up with convincing explanations/ arguments. Still, there is scope to improve the discussion part by elaborating the effect/role of flavonoid /surfactant/ composition solute/solvent for the studied properties of systems considered in this work.

(iv) Discussion and Conclusions should be elaborated further along with a comparison of results for similar surfactants and efforts to be made towards evidencing major gaps in both existing experimental work/ theoretical & computational models.

(v) Further justification/ explanation is essential for most of the claims about the hydrophobic-hydrophilic, hydrophilic-hydrophilic, Van der Waals and hydrogen, bonding, and other non-covalent interactions are mainly based on derivatives and double derivatives of derived data associated with substantial uncertainties.

Author's Response to Decision Letter for (RSOS-210758.R0)

See Appendix A.

Decision letter (RSOS-210758.R1)

Dear Dr Bhattarai:

Title: Physicochemical and spectroscopic investigation of flavonoid dispersed CnTAB micelles
Manuscript ID: RSOS-210758.R1

It is a pleasure to accept your manuscript in its current form for publication in Royal Society Open Science. The chemistry content of Royal Society Open Science is published in collaboration with the Royal Society of Chemistry.

Yours sincerely,
Dr Ellis Wilde
Publishing Editor, Journals

On behalf of the Subject Editor Professor Anthony Stace and the Associate Editor Dr Debashree Ghosh.

RSC Associate Editor

Comments to the Author:

The authors have addressed all the issues raised by the referees and therefore, the paper may be accepted.

Reviewer(s)' Comments to Author:

Appendix A

Journal: Royal Society Open Science

Title: Physicochemical and spectroscopic investigation of flavonoid dispersed CnTAB micelles

Response to Reviewer 1:

- In addition to references 9, 10, use these also

J. Mol. Liq. 310 (2020) 113224

J. Mol. Liq. 293 (2019) 111475

The suggested references are added in the revised manuscript (Pl. See Refs. [11] & [12]).

1. Insert a list of abbreviation used

Abbreviation list is included in the revised manuscript (Pl. see on page 14).

2. Add one or two Keywords in the revised text.

As per you suggestion, keywords are revised now.

3. The authors should highlight the importance of their work in the introduction.

Thank you for your comment. Importance of the work is mentioned in introduction.

4. The length of the manuscript is somehow sufficient. In my opinion, it should more elaborate the introduction part to meet the required standard.

Now, introduction part is elaborated to meet the required standard.

5. What is the novelty of this work, explain nicely.

Required details are provided (Pl. see on page 3).

6 All the equation numbers must be mentioned in the revised text.

We have been incorporated corrections into the revised manuscript file.

7 Give only exact findings from the results and discussion in the conclusion section.

Correction has been done in conclusion section.

8. Reference section must be taken into consideration. Check the references and correct them according to the journal style and format.

References are checked and corrected according to journal's style.

9. English language needs to be improved throughout the manuscript.

Response: English of the manuscript is revised throughout.

Response to Reviewer: 2

(i) Authors are optimistic in the estimation of standard uncertainty in measured and derived properties. Should be re-evaluated. The uncertainty of each measured variable must be given and combined in quadrature to give the uncertainty in the derived properties, which must be stated at a defined confidence interval. The number of significant digits must not exceed one more digit than specified by the estimated uncertainty, especially derived and excess properties. The purity of the sample must be considered in the estimates of uncertainty. Considering the revised uncertainty, all data should be presented in appropriate significant numbers.

Corrections have been done in the revised manuscript.

(ii) Comment on the effect of purity of compound on studied properties of flavonoid dispersed CnTAB micelles.

Corrections have been done in the revised manuscript (Pl. see Table 1).

(iii) Authors focused more on explaining the trends observed rather than coming up with convincing explanations/ arguments. Still, there is scope to improve the discussion part by elaborating the effect/role of flavonoid /surfactant/ composition solute/solvent for the studied properties of systems considered in this work.

Now, discussion part is elaborated, significantly, in the revised manuscript.

(iv) Discussion and Conclusions should be elaborated further along with a comparison of results for similar surfactants and efforts to be made towards evidencing major gaps in both existing experimental work/ theoretical & computational models.

We have elaborated in discussion and conclusion by doing comparison of results for similar surfactants. Please see on pages 12 & 13.

(v) Further justification/ explanation is essential for most of the claims about the hydrophobic-

hydrophilic, hydrophilic-hydrophilic, Van der Waals and hydrogen, bonding, and other non-covalent interactions are mainly based on derivatives and double derivatives of derived data associated with substantial uncertainties.

We have calculated uncertainty and incorporated in the revised manuscript file.